# Postoperative Alanine Aminotransferase Levels Are Associated with Outcomes in Pediatric Patients Undergoing Total Cavopulmonary Connection

**DOI:** 10.3390/children9091410

**Published:** 2022-09-17

**Authors:** Siyao Chen, Han Wang, Dandong Luo, Chongjian Zhang

**Affiliations:** Department of Cardiac Surgery Intensive Care Unit, Guangdong Provincial People’s Hospital, Guangdong Cardiovascular Institute, Guangdong Academy of Medical Sciences, Guangzhou 510080, China

**Keywords:** alanine aminotransferase, total cavopulmonary connection, outcome, association, postoperative, prediction

## Abstract

Background: This single-center, retrospective study aims to determine the association between alanine aminotransferase (ALT) and outcomes in pediatric patients undergoing total cavopulmonary connection (TCPC). Methods: In total, 256 pediatric patients undergoing TCPC were included and divided into a normal-ALT group and a high-ALT group. Clinical data were collected for comparisons between groups, and risk factors of high postoperative ALT were identified by univariate and multivariate analysis. A ROC analysis of the predictive value of postoperative ALT was conducted. Results: Compared to the normal-ALT group, the members of the high-ALT group were 1.6 years older and had significantly higher preoperative creatinine and direct bilirubin levels. The high-ALT group had increased fluid overload, higher vasoactive inotropic drug scores, and inferior central venous pressure. The short-term outcomes in the high-ALT group were markedly worse: they suffered a longer duration of mechanical ventilation (MV), had a higher ICU and hospital length of stay (LOS), and higher rates of mortality, infection, and reintubation. Prolonged ICU and hospital LOS, longer MV, and reintubation were identified as independent risk factors for high postoperative ALT. Postoperative ALT was of high value in predicting reintubation, MV, ICU LOS, and mortality. Conclusions: Elevated postoperative ALT levels are associated with poor short-term outcomes in pediatric patients undergoing TCPC.

## 1. Introduction

Total cavopulmonary connection (TCPC) is a palliative procedure for patients with complex congenital heart anomalies. As a modified approach to Fontan reconstruction, TCPC can be performed with low risk and provides ideal mid- to long-term survival with superior hydrodynamic characteristics as compared to classic Fontan operations. In the TCPC operation, systemic venous return is directed to the pulmonary arterial circulation, connecting both the superior caval vein and the inferior caval vein to the pulmonary artery [1]. By far, the extracardiac conduit TCPC (eTCPC) is the most widely used.

The clinical outcomes of TCPC have been reported by medical centers all over the world. The incidence of arrhythmias was significantly lower in patients with eTCPC as compared with patients treated by intra-atrial lateral tunnel TCPC [2]. Early extubation following TCPC improves postoperative hemodynamics and outcome, regardless of the preoperative hemodynamic status [3]. eTCPC for patients with apicocaval juxtaposition can be safely performed with a good overall survival rate of 95% at 15 years, and without conduit-related complications [4]. Higher preoperative B-type natriuretic peptide levels were associated with poor clinical outcomes following TCPC [5]. Classic morbidities of the original Fontan procedure, such as tachyarrhythmia and thromboembolism, have been reduced in TCPC. Nevertheless, atrioventricular valve insufficiency, ventricular dysfunction, and liver dysfunction, among others, are still a concern [6]. 

Assay of serum alanine aminotransferase (ALT) is not only used as the primary screening tool for detecting acute liver injury, but also used to study postoperative cardiac injury [7]. Notably, liver dysfunction following TCPC has remained an issue. Liver stiffness increases rapidly following TCPC operation, and is associated with perioperative variations in some liver enzymes [8]. Pediatric patients with persistent liver dysfunction following TCPC suffer longer mechanical ventilation and postoperative hospital stays as compared to those with transient liver dysfunction [9]. A study of the outcomes of preadolescents, adolescents, and adult patients undergoing TCPC showed that long-term survival and ventricular ejection fraction were reduced in the older cohort [10]. Serum ALT levels were positively associated with risk factors for cardiovascular diseases in a Caucasian population [11]. The aspartate aminotransferase to ALT ratio is associated with worse clinical outcomes in patients with acute heart failure [12]. Although TCPC can be performed safely with good survival, there remains an unpredictable risk of adverse outcome. It is crucial that we identify the relevant risk factors associated with adverse outcomes following TCPC. 

Is serum ALT associated with outcome in pediatric patients undergoing TCPC? Can serum ALT levels predict the outcome in TCPC? For the most part, these questions remain unanswered. Thus, in the present study, we aim to investigate the association between serum ALT levels and outcome in pediatric patients undergoing TCPC. Furthermore, we analyze the value of postoperative serum ALT level for the prediction of short-term outcomes in children undergoing TCPC. 

## 2. Materials and Methods

### 2.1. Study Design and Population

The clinical data of pediatric patients undergoing TCPC from our center, Guangdong Cardiovascular Institute, Guangdong Provincial People’s Hospital, Guangzhou, China, were collected between 1 January 2012 and 28 February 2022. Patients aged over 14 years were excluded from the study. To minimize the effect of a history of liver diseases on the association between AST and outcome following TCPC, participants with pre-existing liver diseases, including hepatitis B virus or hepatitis C virus, or abnormal preoperative serum ALT levels, were excluded. In total, the data of 256 pediatric patients were included in the present study; the patients were subsequently divided into a normal-ALT group (*n* = 134) defined as maximal ALT value within 48 h post TCPC < 40 U/L and a high-ALT group (*n* = 122), defined as maximal ALT value within 48 h post TCPC > 40 U/L. 

The procedures performed in this study were in accordance with the 1964 Helsinki declaration. The Research Ethics Committee at Guangdong Provincial People’s Hospital approved this retrospective study with a waiver of informed consent in September 2019 (No. GDREC2019338H). 

### 2.2. Data Collection

Preoperative clinical demographic data, including age, sex, height, weight, pre-existing medical history, and medication information, were retrospectively collected from medical records. Preoperative blood chemistry results, including white blood cell counts, hemoglobin, platelets, ALT, creatinine, total bilirubin, and direct bilirubin levels, were collected. Intraoperative clinical data, such as the time of cardiopulmonary bypass and aortic cross clamping, blood transfusion, and TCPC operation including eTCPC and TCPC fenestration, were recorded by an observer blinded to the ALT results. Comparisons of clinical characteristics between the normal-ALT group and high-ALT group were conducted.

Postoperative data, including maximal ALT value 48 h post-TCPC, circulatory fluid load in the first 24 h post-TCPC, and vasoactive inotropic score (VIS) on the first day post-TCPC were recorded. Moreover, the diastolic arterial pressure and inferior central venous pressure (ICVP) in the first hour following TCPC were recorded. The short-term outcome variables within 30 days following TCPC included the hospital length of stay (LOS) (in days), duration of mechanical ventilation (MV) (in hours), rate of all-cause infection, length of ICU stay (in hours), and occurrence of reintubation. Univariate and multivariate logistic analyses of risk factors associated with high ALT following TCPC operation were later conducted to identify the relevant independent risk factors. Furthermore, ROC analyses of the value of ALT in predicting outcomes were conducted.

### 2.3. Statistical Analysis

The Kolmogorov–Smirnov test was used for normality testing. Continuous variables that conform to normal distribution were expressed as mean ± standard deviation, continuous variables that did not conform to the normal distribution were represented as M (Q1, Q3), and count data were expressed in frequency (%). Differences between groups were compared. The variables analyzed by univariate analysis with statistical significances were included in a stepwise multivariate logistic regression analysis. Subsequently, the statistically significant variables in the multivariate analysis were analyzed on the receiver operating characteristic (ROC) curve. Statistical software including SPSS 25.0 and MedCalc 19.0 were used for data processing. *p* < 0.05 was considered statistically significant.

## 3. Results

### 3.1. Clinical Characteristics of Subjects Stratified by Maximal ALT Value within 48 h Post-TCPC

The clinical characteristics of the study population, stratified by maximal ALT value within 48 h post-TCPC, are shown in Table 1. Patients in the high-ALT group were older (*p* < 0.05) and had significantly higher preoperative serum levels of creatinine (*p* < 0.05) and direct bilirubin (*p* < 0.05) as compared to the normal-ALT group (Table 1). During the TCPC operation, pediatric patients in the high-ALT group had significantly more transfusion of RBC than the normal-ALT group (*p* < 0.01). Postoperatively, the high-ALT group had significantly more fluid overload (*p* < 0.01) and higher VIS (*p* < 0.01) within the 24 h following TCPC, along with higher ICVP within 1 h post-TCPC (*p* < 0.01) as compared to the normal ALT group. The variables related to clinical outcomes in the high-ALT group were markedly worse than the normal-ALT group. Compared with the normal-ALT group, the high-ALT group had a longer duration of mechanical ventilation (*p* < 0.01), and a longer ICU stay (*p* < 0.01) and hospital LOS (*p* < 0.01). In addition, pediatric patients in the high-ALT group suffered a markedly higher percentage of mortality (*p* < 0.05), and more all-cause infection (*p* < 0.05) and reintubation (*p* < 0.01), as compared to the normal-ALT group (Table 1).

### 3.2. Univariate and Multivariate Logistic Analyses of Risk Factors Associated with High ALT Following TCPC Operation

To identify the risk factors affecting postoperative ALT, univariate analyses were conducted. Risk factors associated with high postoperative ALT levels were longer LOS (*p* < 0.001, OR 2.992, 95% CI 1.635–6.477), longer duration of mechanical ventilation (*p* < 0.001, OR 7.56, 95% CI 3.494–16.355), longer ICU LOS (*p* < 0.001, OR 7.146, 95% CI 3.641–14.039), need for reintubation (*p* = 0.001, OR 12.941, 95% CI 2.957–56.642) and infection (*p* = 0.012, OR 3.454, 95% CI 1.135–9.078) in the univariate analyses (Table 2). Age, preoperative creatinine and direct bilirubin, blood transfusion, postoperative fluid load, and VIS within 24 h following TCPC, in addition to higher ICVP within 1 h post-TCPC, were adjusted for the multivariate model. Longer ICU LOS (*p* < 0.001, OR 4.911, 95% CI 2.223–10.849), longer hospital LOS (*p* = 0.001, OR 3.235, 95% CI 1.606–6.516), longer duration of mechanical ventilation (*p* = 0.025, OR 2.879, 95% CI 1.143–7.253), and need for reintubation (*p* = 0.003, OR 11.317, 95% CI 2.339–54.758) were identified as independent risk factors for high ALT following TCPC in the multivariate logistic analyses.

### 3.3. ROC Analysis of Postoperative ALT in Predicting Reintubation or All-Cause Infection for Pediatric Patients Undergoing TCPC Operation

To investigate whether postoperative ALT level could predict complications following TCPC, ROC curves of postoperative ALT value for predicting different events following TCPC operation, including reintubation and all-cause infection, were plotted (Figure 1). Postoperative ALT level in pediatric patients undergoing TCPC had a sensitivity of 90.91% and a specificity of 56.41% for predicting reintubation at a cutoff point of 40 U/L. When the postoperative ALT level was at a cutoff point of 40 U/L, the sensitivity and specificity for predicting all-cause infection were 73.91% and 54.94% (Table 3). 

### 3.4. ROC Analysis of Postoperative ALT in Predicting Clinical Outcomes for Pediatric Patients Undergoing TCPC Operation

To investigate whether the postoperative ALT level could predict outcomes following TCPC, ROC curves of postoperative ALT for predicting different outcomes including mortality, and hospital and ICU LOS, as well as MV, were plotted (Figure 2). Postoperative ALT value had a sensitivity of 83.33% and a specificity of 76% for predicting mortality at a cutoff point of 305 U/L. Postoperative ALT level had a sensitivity of 75.76% and a specificity of 74.74% for predicting ICU LOS at a cutoff point of 76 U/L. Postoperative ALT level showed a sensitivity of 69.35% and specificity of 57.73% for predicting hospital LOS at a cutoff point of 37 U/L. When postoperative ALT level was at a cutoff point of 76 U/L, the sensitivity and specificity for predicting MV were 78.85% and 72.06% (Table 3).

## 4. Discussion

Although clinical outcomes after TCPC operation have improved over the years globally, patients with TCPC still face a high risk of intermediate-term cardiovascular events [13]. Moreover, it was reported that adverse outcome-free survival rates 10, 20, and 25 years post-TCPC were 89%, 60%, and 24%, respectively. Additionally, extracardiac TCPC was identified as an independent risk factor for adverse outcomes in a multivariate model [14]. Risk factors affecting long- and short-term outcomes after TCPC include, but are not limited to, the timing of TCPC operation [15], flow dynamics of bilateral superior cavopulmonary shunts [16], older age at stage-2 palliation [17], and systemic to pulmonary collateral blood flow [18]. So far, no studies regarding the relationship between postoperative ALT levels and outcomes following TCPC have been reported. However, at our institution, we have observed that high postoperative ALT levels are often associated with adverse short-term outcomes in pediatric patients undergoing TCPC. Thus, the present study has been designed to test the hypothesis that postoperative serum ALT levels are associated with outcomes following TCPC. We found that pediatric patients with high postoperative ALT levels had worse short-term outcomes, in that they suffered a longer duration of MV and prolonged ICU and hospital LOS, in addition to higher rate of mortality, infection, and reintubation. Prolonged ICU and hospital LOS, longer MV, and a need for reintubation were identified as independent risk factors for high postoperative serum ALT levels. A postoperative serum ALT level > 40 U/L was highly associated with the need for reintubation; a postoperative serum ALT level > 76 U/L was highly associated with prolonged MV and ICU LOS; and ALT > 305 U/L strongly predicted mortality. Of note, the perioperative parameters predicting TCPC outcome remain unclarified. Our study implies the potential of postoperative ALT level as a candidate biomarker discriminating the short-term outcome post-TCPC.

ALT is an enzyme mainly synthesized by the liver, and the serum level of ALT is monitored in clinical practice, serving as a biomarker of hepatic diseases [19]. Moreover, abnormal ALT levels are associated not only with chronic and acute liver diseases, but also with cardiovascular diseases, diabetes, and prognosis post-operation, among others. Many factors could affect the level of serum ALT, including but not limited to gender, age, and ethnicity. A recent study demonstrated that African Americans had significantly lower serum ALT compared to non-African Americans [20]. Older patients with diabetes and ALT levels ≤ 18.5 IU/L often have low muscle strength, independent of any associated metabolic disorders [21]. The AST/ALT ratio predicts overall survival after surgery in gastric cancer patients [22]. Increased ratios of AST/ALT both preoperatively and postoperatively were associated with an increased incidence of postoperative acute renal injury after cardiac surgery [23]. However, previous studies did not demonstrate associations between postoperative ALT levels and outcomes in pediatric patients undergoing TCPC. In the preliminary study conducted at our cardiovascular institute, we observed that postoperative serum ALT levels were higher in patients with a poor outcome following TCPC as compared to those with a good outcome. The preliminary study was based on a small sample. In order to confirm our hypothesis, the present study retrospectively analyzed clinical data involving 256 subjects, confirming our hypothesis that increased postoperative ALT levels are associated with poor outcome in pediatric patients undergoing TCPC. Since this is a single-center experience, the limited number of patients enrolled could obscure the power of our conclusion. Thus, it would be beneficial to include patients from other centers around the world to confirm this conclusion, which we would seek to realize in future study.

Although the present study confirms the association between an increased postoperative serum ALT level and poor postoperative outcome in pediatric patients with the TCPC operation, the potential mechanisms remain unclear. Many factors, including acute and chronic liver injuries, metabolic diseases including diabetes mellitus, ethnicity, age, gender, and cardiovascular diseases, could affect serum ALT levels. Moreover, although patients with higher preoperative creatinine and bilirubin levels and liver diseases were excluded, it is possible that the ALT levels in patients with some degree of preoperatory liver dysfunction enrolled in this study could be within the normal range. The enrollment of these subjects could mitigate the power of our conclusion. We speculate that, in the present study, heavier fluid overload, increased VIS, and increased ICVP after TCPC operation could induce right heart failure and subsequent liver injury, thus resulting in elevated serum ALT levels, which requires further investigation. Additionally, the varied degree of preoperatory ventricular dysfunction could have an impact on the high postoperative ALT levels in patients with complex congenital heart diseases, which would require further investigation and more data to verify.

In conclusion, to our knowledge, this is the first study revealing that elevated postoperative serum ALT levels are strongly associated with a poor short-term outcome in pediatric patients after the TCPC operation. In particular, a postoperative serum ALT level > 40 U/L was of high value in predicting reintubation; >76 U/L was highly associated with prolonged MV and ICU LOS; and ALT > 305 U/L strongly predicted mortality in the short term after TCPC operation. In future, large-sample, long-term multi-center studies are needed to verify the predictive value of serum ALT levels among pediatric patients undergoing TCPC. Our results also imply that postoperative ALT levels could serve as a potential biomarker uniquely predicting the short-term, postoperative outcomes of pediatric patients undergoing TCPC surgery.

## Figures and Tables

**Figure 1 children-09-01410-f001:**
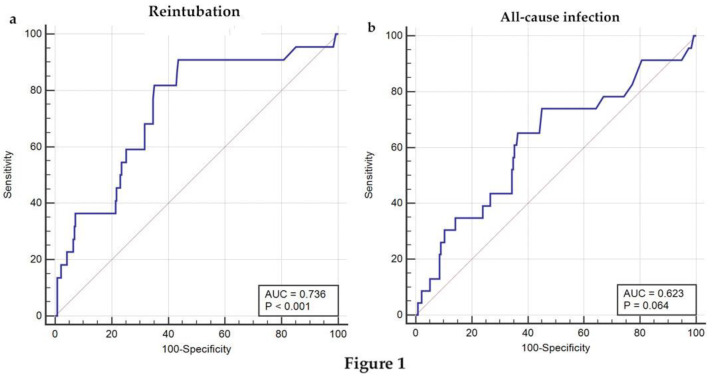
Receiver operating characteristic (ROC) analysis of postoperative ALT in predicting reintubation or all-cause infection for pediatric patients undergoing TCPC. (**a**) ROC curves of postoperative ALT in predicting reintubation; (**b**) ROC curves of postoperative ALT in predicting all-cause infection for pediatric patients post-TCPC operation.

**Figure 2 children-09-01410-f002:**
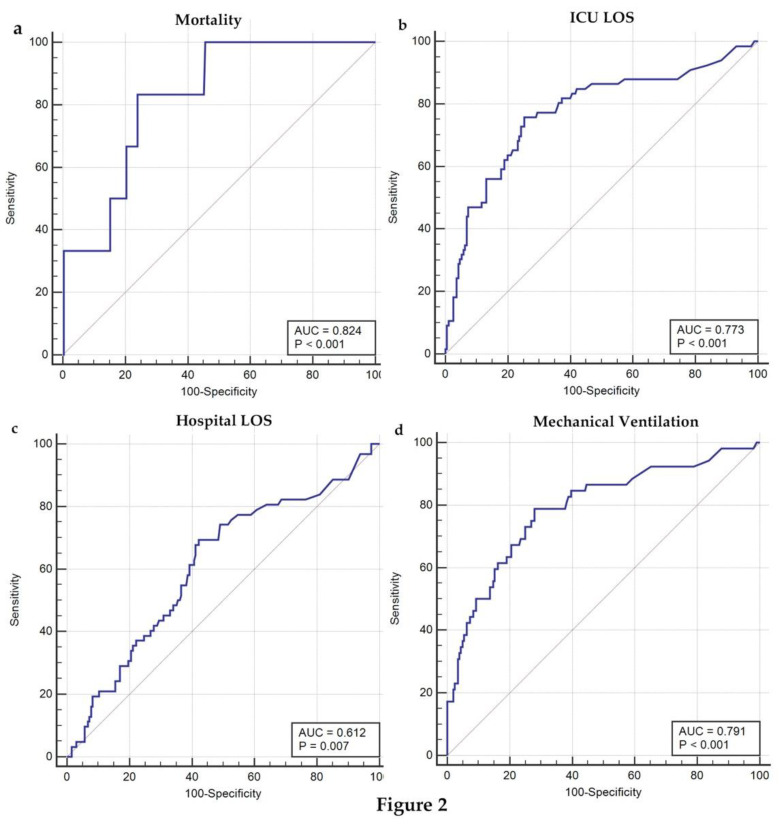
Receiver operating characteristic (ROC) analysis of postoperative ALT in predicting clinical outcomes for pediatric patients undergoing TCPC. (**a**) ROC curves of postoperative ALT in predicting mortality; (**b**) ROC curves of postoperative ALT in predicting postoperative ICU LOS; (**c**) ROC curves of postoperative ALT in predicting postoperative hospital LOS; (**d**) ROC curves of postoperative ALT in predicting duration of mechanical ventilation for pediatric patients undergoing TCPC operation.

**Table 1 children-09-01410-t001:** Clinical characteristics of subjects stratified by maximal ALT value within 48 h post TCPC.

Variables	Normal-ALT Group (*n* = 134)	High-ALT Group (*n* = 122)	*p*-Value
Male (%)	88 (65.7%)	87 (71.3%)	0.404
Female (%)	46 (34.3%)	35 (28.7%)	0.404
Age (years)	4.7 [3.7; 7.7]	6.3 [3.9; 11.8]	0.019 *
Height (cm)	105.0 [97.0; 122.0]	109.5 [97.0; 143.0]	0.179
Weight (kg)	16.0 [14.0; 20.0]	17.0 [14.0; 32.0]	0.085
Preoperative WBC (10⁹/L)	7.9 [6.6; 9.3]	7.9 [6.4; 10.0]	0.809
Preoperative hemoglobin (g/L)	165.0 [154.0; 180.0]	167.0 [154.0; 183.0]	0.497
Preoperative platelet (10⁹/L)	253.0 [208.0; 316.0]	233.0 [189.0; 301.0]	0.067
Preoperative creatinine (μmol/L)	33.0 [26.9; 44.0]	37.0 [30.0; 50.0]	0.018 *
Preoperative ALT (U/L)	15.0 [12.0; 19.0]	15.5 [13.0; 19.0]	0.536
Preoperative total bilirubin (μmol/L)	13.0 [9.2; 16.9]	12.8 [10.3; 17.8]	0.528
Preoperative direct bilirubin (μmol/L)	3.7 [2.7; 4.8]	4.3 [3.0; 5.8]	0.035 *
Preoperative fibrinogen (mg/dL)	2.6 [2.3; 3.2]	2.7 [2.4; 3.3]	0.059
Cardiopulmonary bypass time (h)	118.5 [88.0; 142.0]	123.5 [91.0; 163.0]	0.099
Aortic cross clamp time (h)	47.0 [0.0; 67.0]	48.5 [0.0; 84.0]	0.516
Transfusion of RBC (U)	2.0 [1.0; 3.0]	3.0 [2.0; 6.0]	<0.001 **
eTCPC (%)	49 (36.6%)	47 (38.5%)	0.846
TCPC fenestration (%)	17 (12.7%)	21 (17.2%)	0.4
Maximal ALT within 48 h post-TCPC (U/L)	20.0 [16.0; 25.0]	379.0 [125.0; 1749.0]	<0.001 **
Circulatory fluid load 24 h post-TCPC (mL)	104.8 [–91.0; 397.3]	495.5 [20.0; 1170.0]	<0.001 **
Vasoactive inotropic score 24 h post-TCPC	5.0 [4.0; 8.0]	8.0 [5.0;15.0]	<0.001 **
Postoperative diastolic arterial pressure within 1 h post-TCPC (mmHg)	58.0 [50.0; 65.0]	55.5 [49.0; 62.0]	0.251
Postoperative ICVP 1 h post-TCPC (mmHg)	16.0 [13.0; 18.5]	17.0 [15.0; 20.0]	0.005 **
Postoperative Hospital LOS (d)	14.0 [11.0; 22.0]	22.0 [16.0; 35.0]	<0.001 **
Postoperative mechanical ventilation (h)	5.0 [4.0; 8.0]	16.0 [6.0; 80.0]	<0.001 **
Postoperative ICU LOS (d)	44.0 [24.0; 86.0]	109.0 [46.0; 165.0]	<0.001 **
Postoperative mortality (%)	0 (0.0%)	6 (4.9%)	0.029 *
Postoperative all-cause infection (%)	6 (4.5%)	17 (13.9%)	0.015 *
Postoperative reintubation (%)	2 (1.5%)	20 (16.4%)	<0.001 **

* Significant at *p* < 0.05 level; ** Significant at *p* < 0.01 level; ALT, alanine aminotransferase; TCPC, total cavopulmonary connection; eTCPC, the extracardiac conduit TCPC; ICVP, inferior central venous pressure; LOS, length of stay.

**Table 2 children-09-01410-t002:** Univariate and multivariate logistic analyses of risk factors associated with high ALT following TCPC operation.

Variables	Univariate Analyses			Multivariate Analyses		
	Crude OR	95% CI	*p*-Value	Adjusted OR	95% CI	*p*-Value
Hospital LOS	2.992	1.635–5.477	<0.001 **	3.235	1.606–6.516	0.001 **
Mechanical ventilation	7.56	3.494–16.355	<0.001 **	2.879	1.143–7.253	0.025 *
ICU LOS	7.149	3.641–14.039	<0.001 **	4.911	2.223–10.849	<0.001 **
Reintubation	12.941	2.957–56.642	0.001 **	11.317	2.339–54.758	0.003 **
All-cause infection	3.454	1.315–9.078	0.012 *	2.393	0.789–7.255	0.123

* Significant at *p* < 0.05 level; ** Significant at *p* < 0.01 level; ALT, alanine aminotransferase; TCPC, total cavopulmonary connection; LOS, length of stay.

**Table 3 children-09-01410-t003:** Receiver operating characteristic (ROC) analysis of postoperative ALT in predicting different events for pediatric patients undergoing TCPC.

	AUC	Cutoff Point	Sensitivity	Specificity	95% CI	Z Statistic	*p*-Value	
Mechanical ventilation	0.791	76	78.85	72.06	0.736–0.839	7.667	<0.0001	*1.606–6.516*
ICU LOS	0.773	76	75.76	74.74	0.717–0.823	7.476	<0.0001	*1.143–7.253*
Hospital LOS	0.612	37	69.35	57.73	0.550–0.672	2.702	0.0069	*2.223–10.849*
Reintubation	0.623	40	73.91	54.94	0.560–0.682	1.852	0.0641	*2.339–54.758*
All-cause infection	0.736	40	90.91	56.41	0.678–0.789	4.174	<0.0001	*0.789–7.255*
Mortality	0.824	305	83.33	76	0.771–0.868	4.588	<0.0001	

ALT, alanine aminotransferase; TCPC, total cavopulmonary connection; LOS, length of stay.

## Data Availability

The data used in this study are available upon reasonable request from the corresponding author.

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
