# Peer review of "Postoperative Alanine Aminotransferase Levels Are Associated with Outcomes in Pediatric Patients Undergoing Total Cavopulmonary Connection"

_children, 2022, doi:10.3390/children9091410_

Round 1
Reviewer 1 Report
Thank you for the opportunity to review thiw paper. The aim of this study to reveal the role of the high postoperative ALT levels as a biomarker of short-term postoperative outcome in pediatric patients undergoing to TCPC surgery is confirmed by the results.
1. These patients had higher preoperative creatinine and bilirubin levels and patients with liver diseases were excluded. Based on the bibliography, please comment in the discussion the posibility that some patients had a preoperatory some degree of liver dysfunction with ALT levels in the normal range ( in the context of a complex congenital heart disease).
2. You have mentioned as apossible explanation that a heavier fluid overload, the increased VIS and incrased ICVP after TCPC operation could induce right heart failure and subsequent liver injury postoperatory. Is there any relation with the complexity of the congenital heart disease in the group with high ALT postoperatory and the degree of the preoperatory ventricular dysfunction?
Author Response
1. These patients had higher preoperative creatinine and bilirubin levels and patients with liver diseases were excluded. Based on the bibliography, please comment in the discussion the posibility that some patients had a preoperatory some degree of liver dysfunction with ALT levels in the normal range ( in the context of a complex congenital heart disease).
Response: Dear reviewer, thank you for your excellent suggestions. We agree with you that we should comment in discussion “the possibility that some patients had a preoperatory some degree of liver dysfunction with ALT levels in the normal range”, which we have added in our discussion using the “Track Changes” function for your view.
- You have mentioned as a possible explanation that a heavier fluid overload, the increased VIS and incrased ICVP after TCPC operation could induce right heart failure and subsequent liver injury postoperatory. Is there any relation with the complexity of the congenital heart disease in the group with high ALT postoperatory and the degree of the preoperatory ventricular dysfunction?
Response: Dear reviewer, we appreciate your excellent perceptions, which is indeed innovative and helpful in clarifying the present research topic. We agree it is quite possible that the degree of the preoperatory ventricular dysfunction could have an impact on the high postoperative ALT levels in patients with complex congenital heart diseases, which would require further investigation and more data to verify. Thank you for providing with us an innovative angle to explore this topic further in our future research, which we have added in discussion using the “Track Changes” function for your view.
Reviewer 2 Report
The manuscript by Chen et al investigated the alanine aminotransferase levels in pediatric patients undergoing total cavopulmonary connection.
I have major concerns and my feedback is outlined below:
Since its single centre study, the hypothesis that levels of alanine in pediatric patients needs further investigation, more patients, mechanisms underlying the increase in alanine.
Very limited number of patients enrolled.
It would be beneficial to include patients from other centres and worldwide to confirm this hypothesis.
What is meaning of older group in line 16. Since ages are so closely related, how do the authors determine age and segregate it?
Why the authors decided to examine the alanine levels?
This study is exclusive for males. Do you have data for younger females?
Author Response
- Since its single centre study, the hypothesis that levels of alanine in pediatric patients needs further investigation, more patients, mechanisms underlying the increase in alanine.
Very limited number of patients enrolled.
Response: Dear reviewer, thank you for your excellent suggestions. We agree with you that the hypothesis that levels of alanine in pediatric patients needs further investigation, more patients, mechanisms underlying the increase in alanine.
Of note, compared to other cardiovascular centers, our center is the one of top cardiovascular centers, with the fourth largest number of congenital heart surgeries in China. Last year, there were in total 1935 cases of congenital heart operations in our institute.
A single-centered-study in National Cerebral and Cardiovascular Center, Osaka, Japan enrolled 36 patients underwent TCPC conversion between 1991 and 2014, and made their conclusion that owing to its anti-arrhythmic effect and Fontan pathway recruitment effect, TCPC conversion with an extracardiac conduit prevented the natural decline of exercise tolerance that is seen in classic Fontan patients (https://pubmed.ncbi.nlm.nih.gov/31883324/). Also, another single-centered-study in Japan enrolling in total of 38 patients underwent TCPC between 1998 and 2014 reported their findings (https://pubmed.ncbi.nlm.nih.gov/32979374/). Compared to other single-centered studies, our center has enrolled significantly more TCPC patients, in total 256 pediatric patients with TCPC operations were collected over the last 10 years in our institute.
With the development of nationwide prenatal diagnosis of congenital heart diseases all over China, many parents choose to terminate pregnancy with congenital heart diseases, resulting in a significant decline in the number of children with congenital heart diseases over the years. Thus, we believe the population in need of TCPC operation will decline over the years to come.
We have added in our discussion the limitations of this study using the “Track Changes” function for your view.
- It would be beneficial to include patients from other centres and worldwide to confirm this hypothesis.
Response: Dear reviewer, thank you for your excellent suggestions. We agree with you that it would be beneficial to include patients from other centres and worldwide to confirm this hypothesis, which we would seek to promote and realize in our future study. We have added in our discussion the limitations of this study using the “Track Changes” function for your view.
- What is meaning of older group in line 16. Since ages are so closely related, how do the authors determine age and segregate it?
Response: Dear reviewer, thank you for your excellent question and we appreciate it. We agree with you that we should be more discreet with our expression regarding “the older group” in line 16. We have revised the expression that “Compared to the normal ALT group, the high ALT group were 1.6 years older” in the abstract using the “Track Changes” function for your view.
- Why the authors decided to examine the alanine levels?
Response: Dear reviewer, thank you for your excellent question. In the preliminary study conducted in our cardiovascular institute, we have observed a phenomenon that high postoperative ALT levels are often complicated with adverse short-term outcomes in pediatric patients undergoing TCPC. We have found no reports regarding the relationship between postoperative ALT levels and outcomes following TCPC. Thus, the present study has been designed to examine the alanine levels and test our hypothesis that elevated postoperative serum ALT levels are strongly associated with poor short-term outcome in pediatric patients after TCPC operation.
- This study is exclusive for males. Do you have data for younger females?
Response: Dear reviewer, thank you for your excellent questions. The subjects included are not exclusively males. Both males and females are randomly included in the present study. We are sorry for the confusion we caused due to our indiscreet expression in Table 1. Thus we have added the number and percentage of female subjects in Table 1 using the “Track Changes” function for your view.
Round 2
Reviewer 2 Report
The authors have addressed all the concerns and I have no further comments.